# Exploring the phase stability in interpenetrated diamondoid covalent organic frameworks

Sander Borgmans [1], Sven M. J. Rogge [1✉], Juul S. De Vos [1], Pascal Van Der Voort[2] & Veronique Van Speybroeck [1✉]

Soft porous crystals, which are responsive to external stimuli such as temperature, pressure, or gas adsorption, are being extensively investigated for various technological applications. However, while substantial research has been devoted to stimuli-responsive metal-organic frameworks, structural flexibility in 3D covalent organic frameworks (COFs) remains ill-understood, and is almost exclusively found in COFs exhibiting the diamondoid (**dia**) topology. Herein, we systemically investigate how the structural decoration of these 3D **dia** COFs—their specific building blocks and degree of interpenetration—as well as external triggers such as temperature and guest adsorption may promote or suppress their phase transformations, as captured by a collection of 2D free energy landscapes. Together, these provide a comprehensive understanding of the necessary conditions to design flexible diamondoid COFs. This study reveals how their flexibility originates from the balance between steric hindrance and dispersive interactions of the structural decoration, thereby providing insight into how new flexible 3D COFs can be designed.

[1] Center for Molecular Modeling (CMM), Ghent University, Technologiepark-Zwijnaarde 46, 9052 Zwijnaarde, Belgium. [2] Center for Ordered Materials, Organometallics and Catalysis (COMOC), Department of Inorganic and Physical Chemistry, Ghent University, Krijgslaan 281 (S3), 9000 Gent, Belgium. ✉email: Sven.Rogge@UGent.be; Veronique.VanSpeybroeck@UGent.be

Responsive nanoporous materials can undergo structural phase transitions between different crystalline metastable phases under external stimuli such as temperature, pressure, or adsorption[1]. As phase transitions impact the internal pore architecture of these so-called soft porous crystals (SPCs), these materials are highly sought-after for applications in gas and fluid capture, gas separation, heterogeneous catalysis, nanosensing and drug delivery[2–13]. Thanks to extensive experimental efforts, the range of SPCs has expanded appreciably in recent years, especially for the class of metal-organic framework (MOF) materials[14–16]. Taking MIL-47 and MIL-53 as textbook exemplars[17,18], their well-known winerack topology potentially allows for transitions in a well-defined range of external conditions. However, the actual occurrence of phase transitions under a given set of thermodynamic conditions is uniquely determined by the relative stability of its different phases and the free energy barriers between them. This is a general observation for SPCs: while their topology dictates the potential for flexibility, the degree of interpenetration and the building blocks that decorate this topology—or structural decoration—defines whether or not this flexibility comes to expression. To expand the range of possible SPCs, it is, therefore, key to understand to which extent this structural decoration impacts the relative stability of the different metastable phases in an a priori flexible topology. While several studies have sought to investigate the effect of altering the linker or metal ion in SPCs[19–21], the impact of interpenetration, i.e., the occurrence of two or more individual networks catenated with each other[22], on the relative phase stability in SPCs remains ill-understood despite the vast amount of interpenetrated framework structures. In this work, we, therefore, aim to provide computational guidelines to alter the phase stability and hence phase transition behaviour of SPCs by controlling their degree of interpenetration. To this end, we will quantify the impact of interpenetration on the relative stability of four framework materials exhibiting the flexible diamondoid (**dia**) topology and investigate how external triggers alter this stability.

Although recent years have witnessed a strong increase in the number of soft porous MOF crystals, the number of covalent organic frameworks (COFs) with large-amplitude flexibility under external stimuli remains extremely limited[23–28]. While this may seem plausible given that, in contrast to MOFs, COFs are framework materials that are completely built up from strong covalent bonds[29], the few COFs that do exhibit flexibility shed light on how these strong directed bonds may result in 'soft' materials nonetheless. Remarkably, all flexible COFs found to date almost exclusively exhibit the **dia** topology, which is also regularly encountered in flexible MOFs[30,31]. An example of this is COF-300[32,33], for which the **dia** topology, the non-interpenetrated structure, and its building blocks are visualised in Figs. 1, 2a. Similar to the majority of diamondoid frameworks, COF-300 is interpenetrated such that the individual nets are translated along the twofold rotation axis (the [001] crystal axis in Fig. 2b), belonging to the so-called interpenetration class Ia[34]. When viewed along this axis, the material exhibits large square-shaped (SQ) channels, independent of the degree of interpenetration, as visualised in Fig. 2c. Experiments have shown that, besides these SQ channels, the channel network in diamondoid COFs can adopt various sizes and shapes (see Fig. 2c), either due to different synthesis protocols or the presence of guest molecules. This gives rise to smaller square-shaped channels (sq) or rectangular channels (rect), respectively, as reported in Supplementary Table 1 in Supplementary Note 2. Aside from these variations, other, possibly more complex, material distortions can manifest depending on the external stimuli, such as a collective shearing or individual tilting of the tetratopic nodes[35,36]. However, as these additional material distortions have not yet been associated with flexible diamondoid COFs experimentally, and accounting for their contributions to the phase stability would

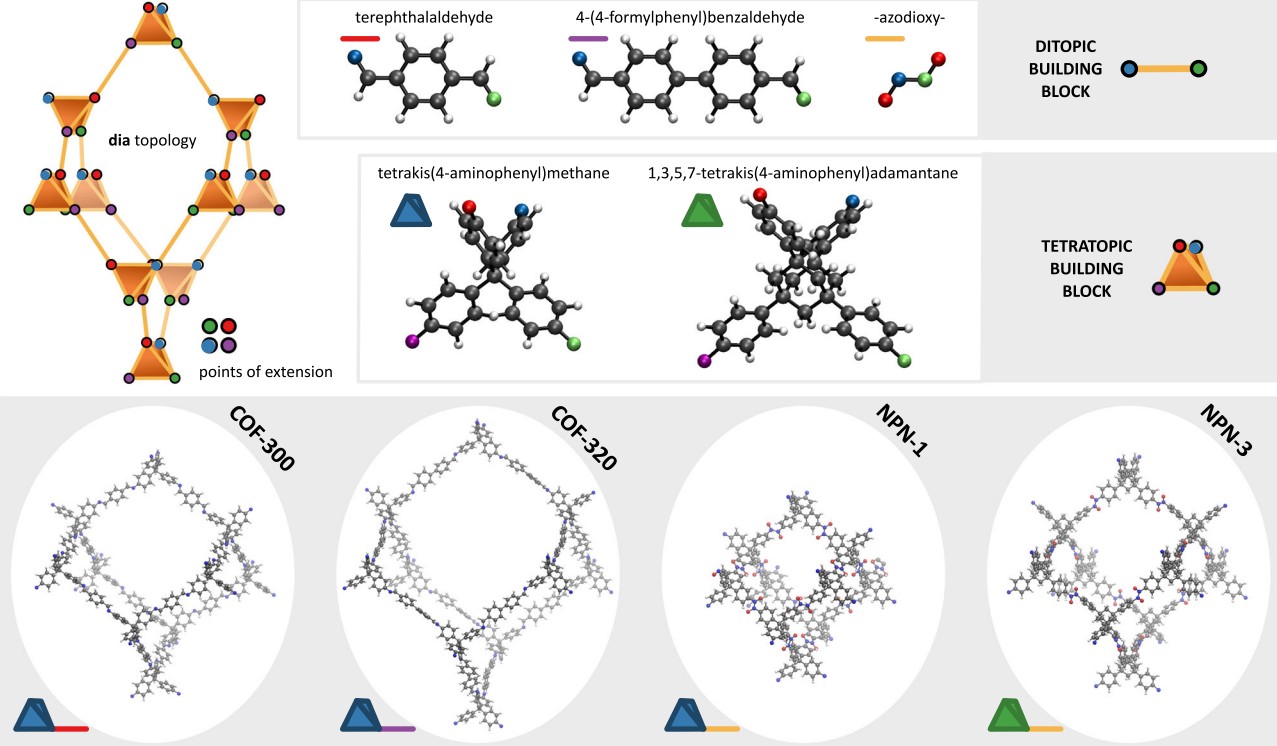

**Fig. 1 Illustration of the four considered materials, and their ditopic and tetratopic building blocks.** As the NPN COFs form an azodioxy linkage through selfcondensation during synthesis, the ditopic building block is labelled as such instead of its precursor. Additionally, the four unique points of extension for each tetratopic building block, coinciding with the nitrogen atoms, have been indicated by the red, green, blue and purple colours.

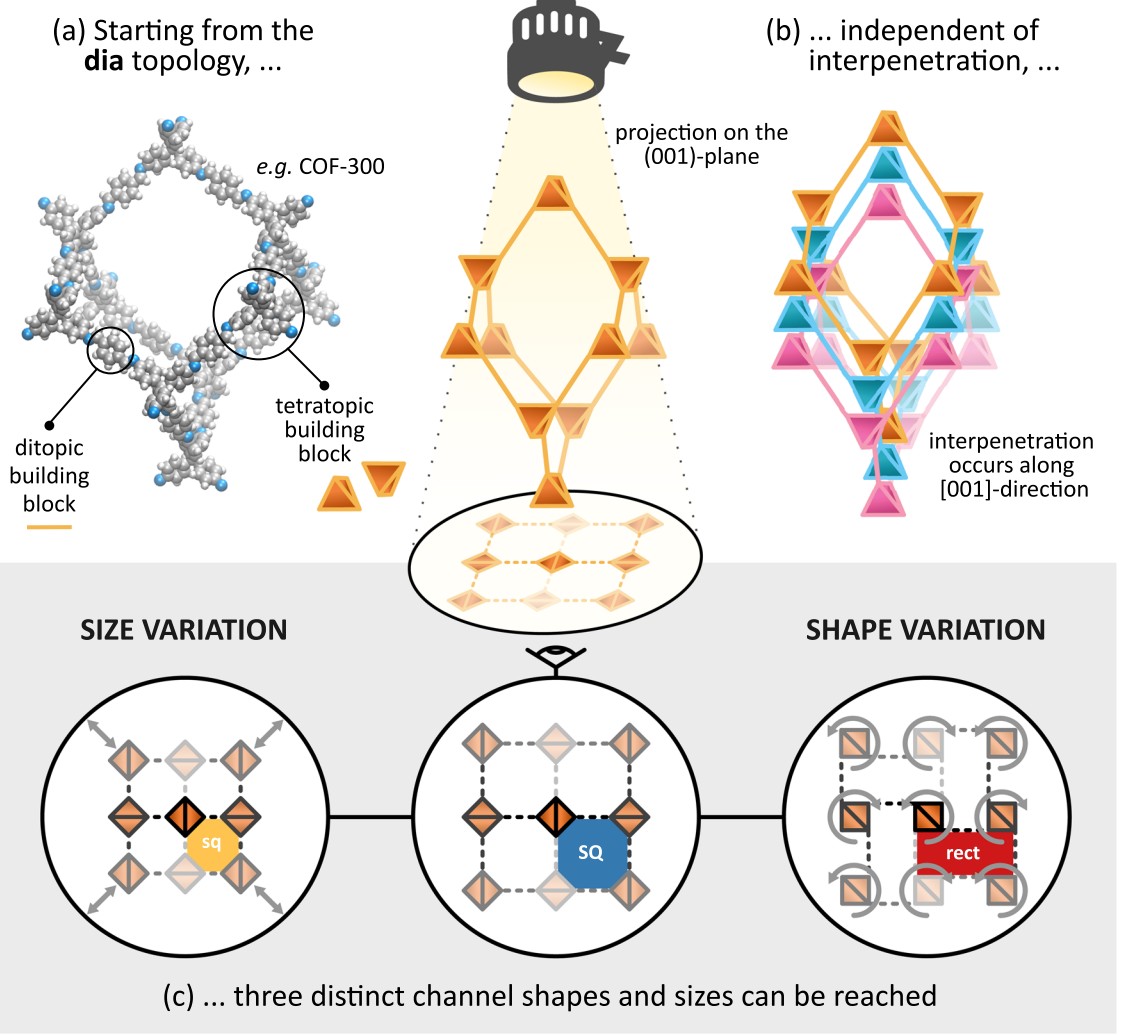

**Fig. 2 Illustration of the dia topology and its experimental channel behaviour.** The dia topology is elaborated by depicting the **a** non-interpenetrated atomic representation of COF-300 and **b** a threefold interpenetrated diamondoid framework. **c** When viewed along the ideal twofold rotation axis ([001]-direction), three distinct channel shapes and sizes emerge, namely, sq, SQ and rect, independent of the degree of interpenetration. These channel shapes and sizes are defined by the relative orientation and distance between adjacent tetratopic building blocks.

require substantially more complex models, they are not explicitly taken into account in this work. As such, in general, three distinct channel shapes can be identified in the diamondoid topology, namely, SQ, sq, and rect. While the observation of different phases at different degrees of interpenetration and experimental conditions alludes to a correlation between them, systematic insight into how the structural decoration and external triggers affect the phase stability in diamondoid COFs is still lacking. Moreover, as evident from the broad variety in the experimental channel shape and size information in Supplementary Table 1, a comprehensive theoretical understanding of the phase behaviour would offer a reliable reference framework for experiments. Therefore, we will herein determine the occurrence and relative stability of these different phases for four **dia** COFs. As illustrated in Fig. 1, these materials represent a versatile set with fundamentally different linkers (imine/azodioxy) and tetratopic building blocks (methane/adamantane). COF-300 and COF-320[37] differ only in the length of their linker, with COF-300 incorporating the shorter linker. Both NPN-1[38] and NPN-3[38] contain a much shorter and more rigid linker; the latter has a more rigid tetratopic building block containing an adamantane cage instead of a tetraphenylmethane moiety. While single-crystal-to-single-

crystal structural transitions have been reported in the literature for COF-300 and COF-320, no such transitions have been reported for either NPN-1 or NPN-3 at the moment (see Supplementary Table 1). In this work, we aim to understand and verify this striking difference, and its connection to their fundamentally different structural decoration. Together, these four **dia** COFs have been observed for a wide range of interpenetration, from four to nine interpenetrating nets, thereby forming a varied set of materials to derive general relations between the interpenetration and flexibility in framework materials exhibiting the **dia** topology.

## Results

**Choice of collective variables to map the phase transition**. From a fundamental point of view, identifying the occurrence of different metastable phases, their transition barrier, and the corresponding transition mechanism, requires knowledge of the free energy of these phases and the pathways connecting them, and this at the thermodynamic conditions and degree of interpenetration of interest. Such a free energy landscape is expressed as a function of so-called collective variables (CVs) that

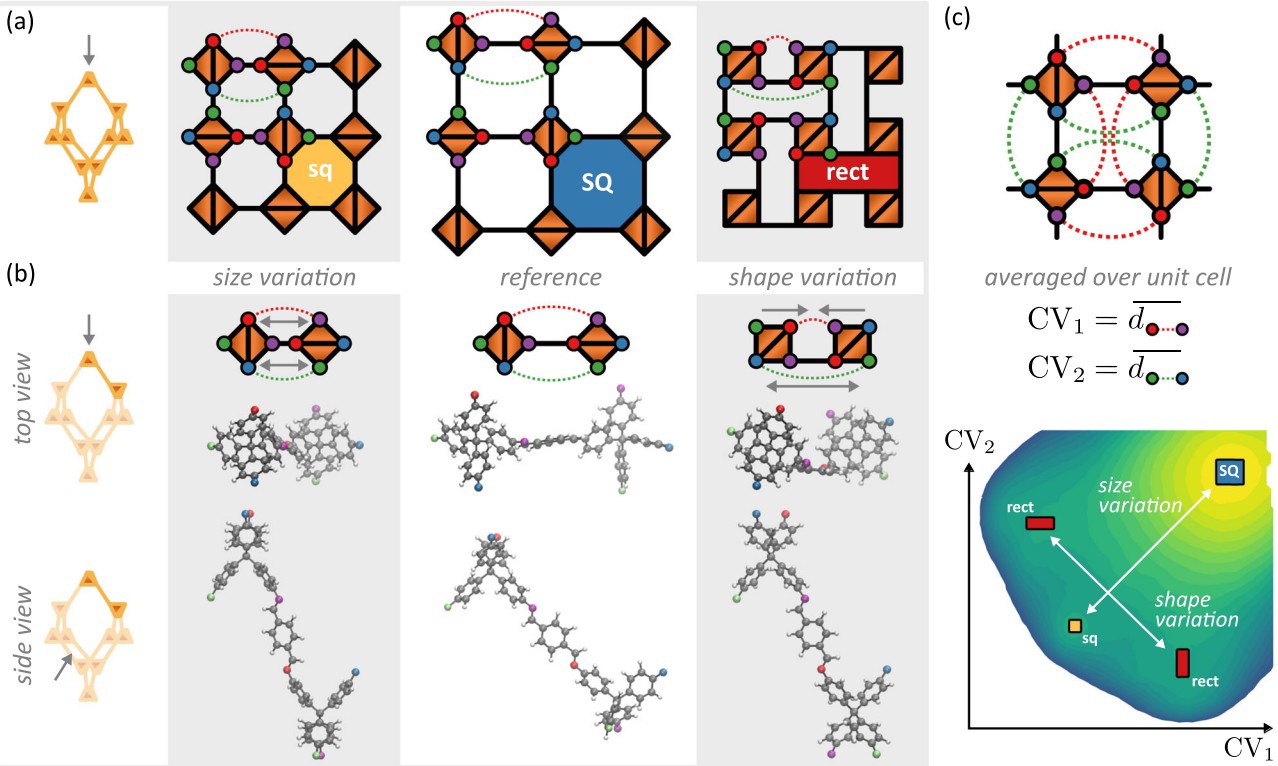

**Fig. 3 Definition of the collective variables and the corresponding transition mechanism between the three distinct phases, SQ, sq and rect. a** The distinct phases, where the points of extensions have been indicated by coloured circles, in correspondence with Fig. 1. **b** The variations in channel size and shape, relative to the reference, can be defined via two types of distances, indicated by the red and green dotted lines. Pure size variations only occur when both distances increase or decrease simultaneously, whereas shape variations are introduced when the distances change differently. Their atomic representations, in both top and side view, for the case of COF-300, are provided to illustrate the corresponding transition mechanism. This clearly visualises that a size decrease elongates the dia cage in the [001] direction while compressing it in the perpendicular directions, while a shape change is facilitated by rotating tetratopic building blocks. **c** Through averaging the two types of distances in panels **a** and **b** over the unit cell, two collective variables are obtained, capable of describing the simultaneous occurrence of all phases and validating the choice of CVs, as demonstrated in the free energy surface for fourfold interpenetrated COF-300.

distinguish between all potential phases. For the three **dia** phases, this means that the CVs should be able to describe both variations in shape and size of the originally square-shaped channels, so as to describe the SQ, sq, and rect phases. From the projection of these three phases on the (001) plane, shown in Fig. 2c, it is clear that such CVs can be constructed from the relative projected positions of the points of extension that connect adjacent tetratopic building blocks in the structure. To see this, Fig. 3a visualises the three **dia** phases for which the four points of extension of the tetratopic building blocks have been indicated with coloured circles, in correspondence to Fig. 1. For two adjacent vertices, shown in Fig. 3b, both shape and size variations can be distinguished based on the two distances indicated with red and green dotted lines. These distances are calculated between projected points of extensions that are not directly connected within the same tetratopic building block. When these distances either shrink or expand simultaneously, they describe a size variation defining the transition between the sq and SQ phases. In contrast, when these distances do not vary identically, they yield an additional shape variation, giving rise to transitions between the SQ and rect phases or between the sq and rect phases. From the atomic representations of each of the transition mechanisms in Fig. 3b, it is clear that size variations elongate or compress the diamondoid cage in the [001] direction, whereas shape changes are facilitated by a complementary rotation of two adjacent tetratopic building blocks. For the whole unit cell, containing four tetratopic building blocks, as elaborated in Supplementary Note 1,

a phase transition requires a concerted motion of all four building units. Therefore, the set of CVs used throughout this work are obtained by considering the two aforementioned distances, averaged over all such distances present in the unit cell. These CVs correlate well with macroscopic system properties, as outlined in Supplementary Note 3 and Supplementary Figs. 1, 2. As shown in Fig. 3c, these CVs succeed in describing the relative stability of the different phases in the fourfold interpenetrated COF-300 at 300 K and will therefore be adopted in the remainder of this work.

**Free energy landscape as a function of structural decoration.** Having established that the two chosen collective variables can distinguish between all **dia** phases, we first aim to investigate how interpenetration alters the relative stability of these phases. Intuitively, an increasing degree of interpenetration implies a stronger steric repulsion as it increases the materials' density and limits the accessible space around the building blocks. However, as we will demonstrate later in this work, interpenetration impacts the stability of the different phases differently, thereby showing the potential to also promote the existence of multiple phases and phase transitions. Hence, interpenetration can fundamentally alter to what extent the COF is responsive to external stimuli such as temperature, pressure, or gas adsorption. Consequently, as the degree of interpenetration may be modified by varying, e.g., the synthesis conditions[39–41], interpenetration

allows for control over the range of thermodynamic conditions for which the material shows flexibility. To outline this effect on the range of possible channel shapes and sizes, the degree of interpenetration is varied for the considered materials from onefold (no interpenetration) to eightfold (or elevenfold for COF-320). The maximal degree of interpenetration corresponds to the point at which the individual nets are so close together that they start to overlap, resulting in prohibitively large steric repulsion.

Focusing on the archetypal COF-300, Fig. 4 showcases that the relative stability of the distinct phases is indeed substantially impacted depending on the interpenetration, which is clear when considering the variations in the free energy landscapes at 300 K. For the non-interpenetrated COF-300, a large accessible phase space is observed with a single broad minimum for a large square-shaped channel (SQ). When introducing a second catenated net, evidently, the accessible phase space decreases due to the steric crowding, which destabilises small channel sizes. However, the dispersive interactions between the nets counteract this, resulting in the emergence of a metastable small square-shaped channel phase (sq). This becomes even more pronounced when adding a third net, where the metastable sq phase is pushed towards a slightly larger channel size, and metastable rectangular shapes (rect) emerge. Ultimately, when adding a fourth net, the balance between steric hindrance and dispersive interactions shifts. This suppresses the manifestation of small square-shaped channels and pushes the metastable state towards a rectangular shape. In this way, the attractive dispersion is maximised while avoiding steric crowding around the building blocks. Similarly, looking at a fivefold interpenetration, the metastable rect phases are pushed towards an even larger size and shape anisotropy, until the metastable phases completely disappear for all degrees of interpenetration higher than five. Further increasing the degree of interpenetration at this point only limits the accessible phase space, driven by the steric hindrance. As such, by adjusting the degree of interpenetration, the size and shape variations of each emerging phase can be governed, thus controlling the compliance of the 1D channel shape, although the SQ phase remains the most stable under these conditions.

Similar free energy landscapes for COF-300 at other temperatures (50, 100 and 400 K), as well as for the remaining materials, namely, COF-320 (at 50, 100, 300 and 400 K), NPN-1 (at 300 K) and NPN-3 (at 300 K), are reported in Supplementary Figs. 6–15 in Supplementary Note 5. In Fig. 5, the main features of these free energy landscapes are summarised for COF-300 and COF-320, as a function of the degree of interpenetration and temperature. It depicts the free energy differences between the available metastable states and the stable SQ state in the top panels, and their corresponding transition barriers in the bottom panels. Notably, no data for the NPN COFs is shown, as our simulations demonstrate that the structural decoration of these materials, especially their short and rigid linker, precludes flexibility. This gives rise to only a single stable SQ phase for NPN-1 and NPN-3 regardless of interpenetration or temperature, despite having the same **dia** topology as the flexible COF-300 and COF-320. Similarly, as the structural decoration of the highly interpenetrated (and the non-interpenetrated) frameworks inhibits the materialisation of more than one phase, this data is omitted as well.

Figure 5 demonstrates that, at low degrees of interpenetration, the metastable sq phase appears for COF-300 and COF-320, irrespective of the temperature. This sq phase later disappears for intermediate degrees of interpenetration in favour of the metastable rect phase, which finally disappears as well at higher degrees of interpenetration, leaving only the SQ phase. Moreover, while the transition process between the SQ phase to any metastable phase is highly activated with free energy barriers exceeding $100 \, \text{kJ mol}^{-1}$, the corresponding transition barriers generally decrease with increasing degree of interpenetration, until steric crowding makes the small channels inaccessible, in line with the increasing Pauli repulsion between the catenated nets. Conversely, no clear trend is observed when considering the reverse transition barrier height, from any metastable phase to the SQ phase, as a function of the interpenetration. Interestingly, Fig. 5 does show significant reverse transition barriers to the SQ phase, going as high as $70 \, \text{kJ mol}^{-1}$ for fivefold interpenetrated COF-320 at 100 K. Evidently, Fig. 5 allows us to extract general

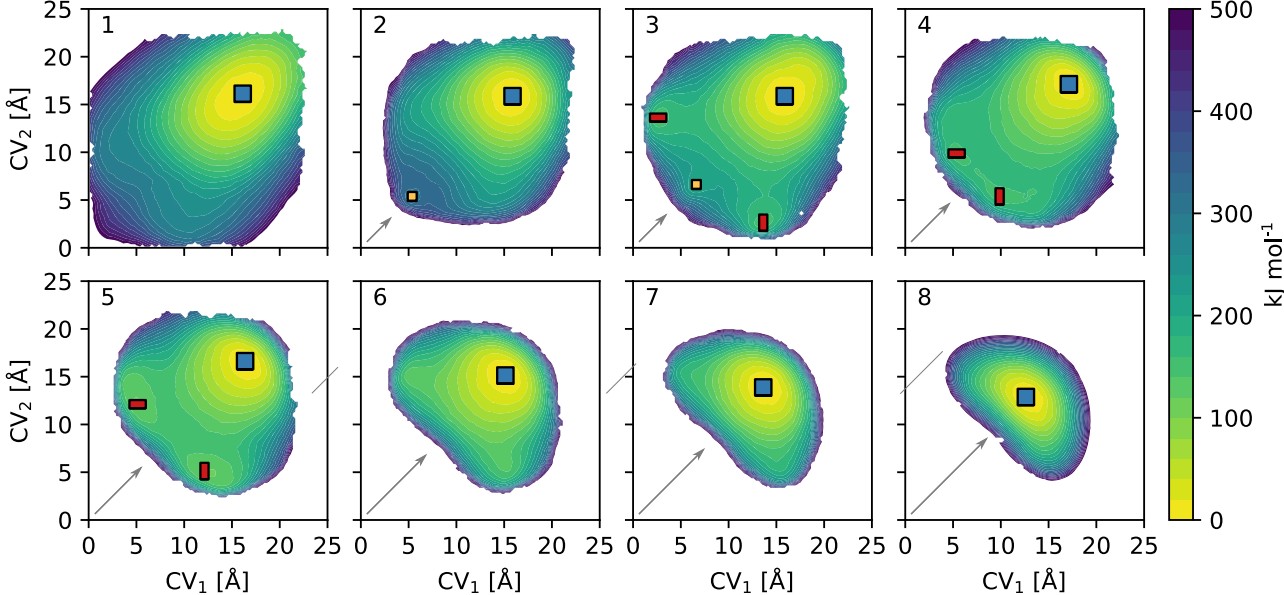

**Fig. 4 Free energy surfaces for COF-300 at 300 K as a function of interpenetration.** The degree of interpenetration varies from one (non-interpenetrated) to eightfold interpenetrated, as indicated in the top left corner of each subpanel. For each subpanel, the different (meta)stable phases are indicated. Regions with free energy exceeding the free energy minimum by more than $500 \, \text{kJ mol}^{-1}$ are omitted from the plot, as they are not accessible under realistic conditions. The arrows indicate the growing steric crowding with increasing interpenetration.

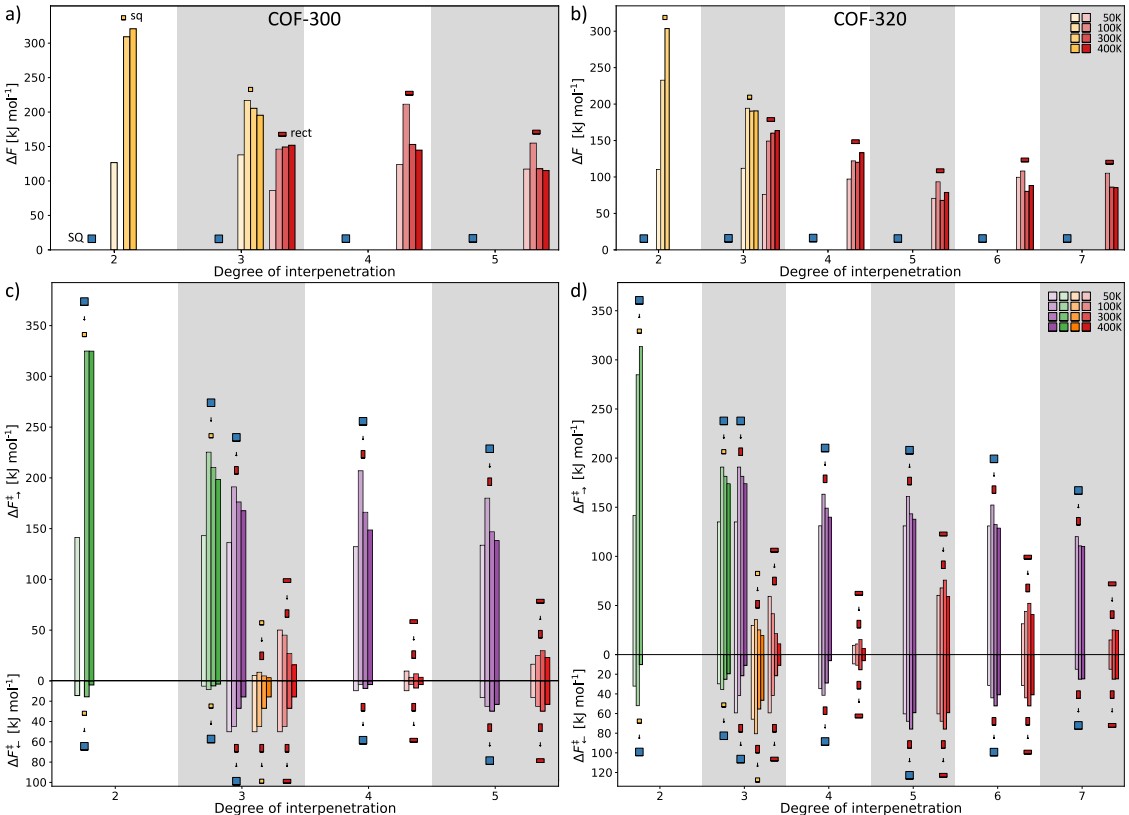

**Fig. 5 Overview of the relative phase stability as a function of the degree of interpenetration and temperature.** The phase stability is visualised through the free energy difference of each (meta)stable phase with respect to the most stable SQ phase for **a** COF-300 and **b** COF-320. Additionally, to illustrate the stability of each phase, the transition barrier height between them is illustrated as well, for **c** COF-300 and **d** COF-320. The temperature effect is visualised through the colour shade, going from 50 K (light) to 400 K (dark), whereas the colour relates to the relevant phase symbols. The different symbols introduced in Fig. 3 are used to distinguish between the different phases.

trends, but it abstractifies the size and shape anisotropy variations that are visible in the free energy landscapes, as depicted in Fig. 4. Importantly, when comparing these variations in Supplementary Figs. 6–15 as a function of the structural decoration, the NPN COFs do exhibit similar stabilisation and destabilisation effects as COF-300 and COF-320, with a very similar shape of the accessible free energy landscape, although the stabilisation effects do not give rise to additional (meta)stable phases.

Finally, while it is clear that the degree of interpenetration and the temperature impart significant variations to the transition barriers, the SQ phase remains the most stable phase for all considered materials. However, as reported in Supplementary Table 1, various channel shapes emerge in diamondoid COFs when exposed to different guest molecules, such as water. As such, we herein sought to explore how the inclusion of water impacts the structure and relative stability of the different phases in COF-300(7), which experimentally amounts to a volume contraction of about 5% for the SQ phase[26–28]. Similar to Fig. 4, the effect of water loading is observed by analysing the variations in the free energy landscapes, when increasing the loading from one to 16 water molecules per channel. This is illustrated in Fig. 6, where the loading is indicated in the top left corner of each pane. From the final panel, it is clear that the SQ channel first decreases in size when water is present in the pores, with a minimum volume for a loading of eight molecules per channel, after which the size increases again. Additionally, a decrease in the available phase space can be observed, due to the density increase, similar to the interpenetration effect. When comparing the most contracted water-filled framework to the guest-free framework, the contraction corresponds to a change (expressed as a

function of the collective variables) from $13.9\,\text{Å} \times 14.1\,\text{Å}$ to $13.1\,\text{Å} \times 13.6\,\text{Å}$, or, expressed in terms of the unit cell volume, from circa $5594\,\text{Å}^3$ to circa $5275\,\text{Å}^3$ (−6%). While these values differ somewhat from those observed experimentally, as tabulated in Supplementary Table 6, the relative change in volume upon water adsorption is very similar to the values from those references mentioned in Supplementary Table 1, succeeding in a qualitative description of the experimentally observed pore shrinkage upon water adsorption. This adsorption-induced shrinkage is substantially smaller than the potential temperature-induced flexibility between the SQ and sq phases for the empty framework predicted in Fig. 4.

## Discussion

The characterisation of responsive nanoporous materials, their internal transition mechanisms, and their corresponding operating range is essential to understand, improve and apply them in an industrial context. Only through the identification of responsive structural decorations, and the relevant triggers that can potentially induce transitions, one can systematically expand the collection of functional SPCs tailored to specific applications. Sparked by a lack of such information for COFs, we herein investigated the effect of different structural decorations—the building blocks and the degree of interpenetration—in the inherently flexible **dia** topology on the potential for phase transformations. To this end, the different experimental phases (sq, SQ and rect) that emerge when viewing the structure along the channel direction were characterised, and their internal transition mechanism was exposed as the cooperative rotation

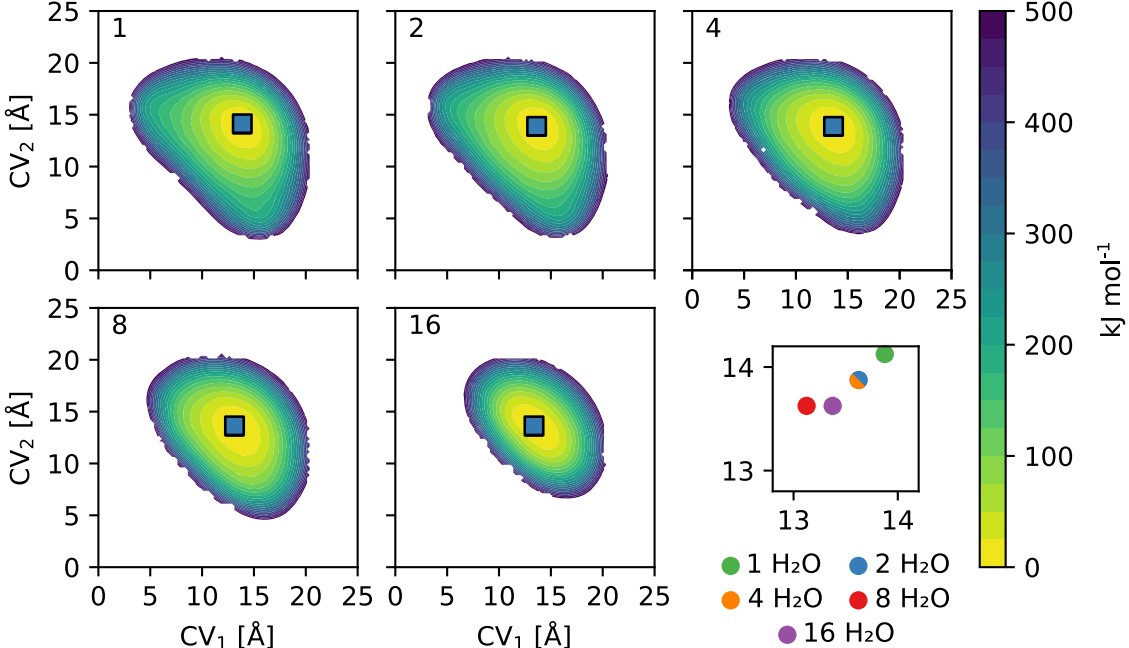

**Fig. 6 Free energy surfaces for COF-300(7) at 300 K as a function of water loading.** The water loading is exponentially varied from one to 16 molecules per channel, as indicated in the top left corner of each subpanel, with four channels per unit cell. The different (meta)stable phases are indicated in the figure. Regions with free energy exceeding the free energy minimum by more than 500 kJ mol$^{-1}$ are omitted from the plot, as they are not accessible under realistic conditions. The final subpanel shows the exact position of the SQ phase for each loading in detail.

and translation of neighbouring tetratopic building blocks perpendicular to the channel direction.

The relative phase stability for four **dia** materials was characterised using two-dimensional free energy landscapes, capturing both size and shape variations of the channels, and their respective transitions. From our enhanced sampling molecular dynamics calculations, clear trends could be observed. While certain structural decorations, such as the rigid azodioxy linker and adamantane cage in the NPN COFs, preclude flexibility, others, such as the degree of interpenetration, can fundamentally shift the delicate equilibrium between the stabilisation and destabilisation effects, driving the phase equilibrium. These (de)stabilisation effects originate from the steric hindrance and dispersive interactions, which disfavour and promote closely packed phases, respectively.

Depending on the number of catenated nets, metastable phases arise in COF-300 and COF-320, stabilised by attractive dispersive interactions between the nets, that become unstable for large degrees of interpenetration due to steric crowding. This gives rise to an optimal, intermediate degree of interpenetration for each metastable phase, with significant transition barriers that could trap the structure in a certain phase. However, by varying the temperature, we observed that the temperature cannot be straightforwardly used as a driving force to completely destabilise the SQ phase. Instead, through introducing water molecules, significant size variations could be induced, shifting the SQ phase towards lower volumes. Finally, regardless of the structural decoration and the considered external triggers, a large square-shaped channel was identified as the most stable phase, warranting more investigation to explore whether building blocks can be designed that destabilise this phase. Our results predict that such a building block should be sufficiently long to allow for stabilisation of the sq or rect phase before steric hindrance destabilises these phases again. Moreover, the here-derived workflow can easily be extended to probe the effect of (anisotropic) stress as a potential trigger towards certain phases,

introducing new application avenues, such as a configurable sensing device with adaptable channels.

## Methods

**Structure generation**. All initial materials were generated in silico using our in-house structure assembly software, which is based on a top-down approach starting from the building blocks of the material and the underlying topology. A detailed procedure of this protocol can be found in Supplementary Note 1, together with a mathematical description of the interpenetrated **dia** topologies.

**Computational details**. The enhanced sampling molecular dynamics (MD) simulations were performed in the ($N$, $P$, $\boldsymbol{\sigma_a} = \mathbf{0}$, $T$) ensemble[42] using the in-house software package Yaff[43] as the MD engine, employing a velocity Verlet integration scheme with a timestep of 0.5 fs. The temperature was fixed at 300 K with a Nosé-Hoover chain thermostat[44–46] with three beads and a relaxation time of 100 fs. The pressure was controlled at 1 bar by a Martyna–Tuckerman–Tobias–Klein barostat[47,48] with a relaxation time of 1000 fs. The interatomic interactions were modelled through system-specific force fields, derived in accordance with the general QuickFF 2.2.4 protocol[49,50], using the default parametrization (see Supplementary Note 6 and Supplementary Figs. 17–21 therein for the force field derivation and validation). The electrostatic interactions were computed with an Ewald summation using a real space cut-off $r_{cut}$ of 15 Å, a scaling factor $\alpha$ equal to 0.213 Å$^{-1}$, and a reciprocal space cut-off $k_{max}$ of 0.320 Å$^{-1}$. Analogously, the van der Waals interactions were calculated with a real space cut-off of 15 Å. Both cut-offs were smoothed using a truncation model. For the water loading simulations, the TIP4P/2005f force field was used[51,52], as elaborated in Supplementary Note 6.5.

To enhance the exploration of the phase space and access all possible phases, umbrella sampling was adopted using a harmonic bias potential as a function of the collective variables as defined in Fig. 3. For the CVs, a uniform grid with a spacing of 1 Å was employed, extended with small patches where a spacing of 0.5 Å was used to accurately reproduce the free energy barriers in regions where neighbouring simulations did not overlap adequately, as reported in Supplementary Tables 2–5. The force constants for the harmonic bias potential ($\kappa_i$) of both CVs were initialised at 10 kJ mol$^{-1}$ Å$^{-2}$, and increased until a maximum of 150 kJ mol$^{-1}$ Å$^{-2}$ for those grid points where the simulation deviated from the umbrella centre (due to a steep free energy profile). Each individual simulation had a run time of 15 ps, including an equilibration time of 5 ps. The resulting free energy from the collection of simulations was calculated with the weighted histogram analysis method (WHAM)[53], using the software package from ref. [54]. Since the WHAM code failed to converge in situations where the sampled two-dimensional phase space was not rectangular, an iterative procedure was applied to combine smaller rectangular regions, as described in Supplementary Figs. 3–5 of Supplementary Note 4. This procedure was validated through the application of

our in-house WHAM code in ThermoLIB, which did not suffer from convergence issues.

The resulting free energy surfaces were symmetrized in accordance with the symmetry of the collective variables, ensuring smoother profiles for the minimal free energy path (MFEP) analysis. The MFEP analysis was performed with the Minimum Energy Path Surface Analysis (MEPSA) code[55], as elaborated in Supplementary Note 5.3. The barrier heights in Fig. 5 for COF-300 and COF-320 were derived from the one-dimensional MFEPs as the difference between the local minimum and the neighbouring local maximum, providing a clear measure of the transition probability, as illustrated in Supplementary Fig. 16.

## Data availability

The relevant input files and computational data which generated the results of this work are available from the online GitHub repository at https://github.com/SanderBorgmans/SupportingInformation or upon request from the corresponding authors.

## Code availability

The Gaussian code, used to perform the ab initio cluster calculations, can be licensed from Gaussian, Inc. (see https://gaussian.com/). The QuickFF (used to derive the force fields), Yaff (MD engine) and ThermoLIB (free energy evaluation engine) software packages are freely accessible via https://molmod.ugent.be/software/. The MEPSA code for the minimal free energy path analysis is freely available at http://bioweb.cbm.uam.es/software/MEPSA/. Representative input and processing scripts are available at https://github.com/SanderBorgmans/SupportingInformation.

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

## Acknowledgements

This work is supported by the Research Board of Ghent University (BOF) through a Concerted Research Action (GOA010-17). S.M.J.R. and J.S.D.V. acknowledge the Fund for Scientific Research–Flanders (FWO) for a postdoctoral fellowship (grant no. 12T3522N) and a strategic basic (SB) research fellowship (grant no. 1S94519N). V.V.S. acknowledges the Research Board of Ghent University (BOF). The computational resources (Stevin Supercomputer Infrastructure) and services used in this work were provided by VSC (Flemish Supercomputer Center), funded by Ghent University, FWO, and the Flemish Government—department EWI.

## Author contributions

S.B., S.M.J.R. and V.V.S. initiated the discussion and designed the paper. S.B., S.M.J.R., J.S.D.V., P.V.D.V. and V.V.S. were involved in the discussion of the results. S.B., S.M.J.R. and V.V.S. wrote the manuscript with the contributions of all authors. J.S.D.V. generated the structures and initial force fields. S.B. extended the force fields where necessary, performed the enhanced sampling calculations and analysed the results.

## Competing interests

The authors declare no competing interests.
