## [Peer Review File · Communications Chemistry]

Reviewers' comments:

Reviewer #1 (Remarks to the Author):

This manuscript reports a computational exploration of the flexibility landscape of a series of diamondoid covalent organic frameworks. The problem of establishing composition—flexibility relationships in this family is approached in an extremely logical manner: the authors select a series of different COFs that share topology but use different node and linker chemistries; they also explore the role of varying degrees of framework interpenetration. Flexibility in these COFs is parameterised in terms of two distortions (referred to here as collective variables). The associated phase space is mapped using molecular dynamics simulations driven by QuickFF-derived force fields, carried out in combination with umbrella sampling. The key results are shown in Figures 4 and 5: the FES is always characterised by a deep minimum at the 'sq' phase – irrespective of chemistry or degree of interpenetration; some small variations in the positions of metastable phases (local minima) are observed.

I am entirely sympathetic to the motivation for this study. It is clearly very well carried out, and the paper is well written (in particular it is beautifully illustrated). Nevertheless I don't think that the study can be published in Communications Chemistry because it fails to reproduce or rationalise the experimental flexibility of e.g. COF-300. I do not see that the key claim of page 9 "...the relative stability of the distinct phases is indeed substantially impacted depending on the interpenetration" is actually supported by the data of Figure 4: the sq phase is always the most stable; the same is true with variation in chemistry or temperature (Figure 5). As such, the study can only have limited impact within its immediate field. Probably the real value of the paper is to set out a methodology for exploring flexibility in diamondoid frameworks – this might be suitable for a society journal, for example. But even in that case it would have been satisfying to see the authors engage with the difference between calculation and experiment to uncover the underlying reason for the discrepancy between the two.

Wherever the authors choose to send on this manuscript, one point I would encourage them to take into account is to tone down their claims regarding the universality of their description of dia phases. There are many additional symmetry-breaking distortions - e.g. shear, loss of edge transitivity, orthogonal tilts - that are not taken into account here.

Reviewer #2 (Remarks to the Author):

This manuscript by Van Speybroeck and co-workers presents the Free Energy Surfaces of four related diamondoid COFs: COF-300, COF-320, NPN-1 and NPN-3 and discusses the availability of structural transitions between large square pore, small square pore and rectangle pore geometries for different degrees of interpenetration, across the four COFs (two nodes, three linkers).

Clearly, the authors have put quite some thought into the presentation of their results, especially their figures, which I commend them for.

My comments are therefore minor.

There are two more things I'd like to see in the SI:

In Table S1, for the filled squares, it would be nice to see the guest molecule(s) named as the extent and type of interaction with the framework would be expected to affect the observed COF structure. The second thing that would be nice is a table in the SI (probably linked to the github) that links the

two CVs (useful to computational colleagues) to lattice vectors and pore size (more recognizable to experimental colleagues) and then the structures themselves on github (I found the initial structures, not the optimized ones)

Typos:

Page 7: There is a reference to "Section ???" - obviously something didn't resolve.

Page 9: "buildin" blocks

Page S-4: "interpenetration" axis

Reviewer #3 (Remarks to the Author):

The paper by Borgmans et al. reports a theoretical investigation of different diamondoid COF networks concerning their flexibility to undergo pressure or temperature-induced phase transformations. The corresponding "breathing" effect for the related crystalline porous frameworks from coordination compounds or MOFs has attracted a lot of interest and research activity. As the authors correctly point out, this is not so for the covalent organic frameworks with their potentially "stiffer" bonds. However, also the 3D COFs based on the dia topology can undergo a transformation leading to reduced cell volumes. Since COFs have their advantages over MOFs in particular applications, such a study is interesting and relevant for the entire field.

The authors have used their QuickFF strategy to train a molecular mechanics potential wrt. to DFT reference data for the particular system. They investigated different COFs, varying the linker length and rigidity. Most importantly, the dia topology is prone to interpenetration and the results were thus compared for different degrees of interpenetration. Free energies for deformation along the two relevant deformations towards an isotropically reduced cell (SQ -> sq) and a deformed system with rectangular pores (SQ -> rect) via a rotation of the 4 connected vertices were computed.

The paper is very well-readable and clear. The viewgraphs are exceptionally well done and help in understanding the approach. I also find the definition and discussion of the collective variables for the above-mentioned deformations of the MOFs smart and easy to follow.

Overall, this is a very interesting paper and I recommend its publication as is (on page 7 the authors should fix the missing reference (Section ???)).

REVIEWER #1

This manuscript reports a computational exploration of the flexibility landscape of a series of diamondoid covalent organic frameworks. The problem of establishing composition—flexibility relationships in this family is approached in an extremely logical manner: the authors select a series of different COFs that share topology but use different node and linker chemistries; they also explore the role of varying degrees of framework interpenetration. Flexibility in these COFs is parameterised in terms of two distortions (referred to here as collective variables). The associated phase space is mapped using molecular dynamics simulations driven by QuickFF-derived force fields, carried out in combination with umbrella sampling. The key results are shown in Figures 4 and 5: the FES is always characterised by a deep minimum at the 'sq' phase – irrespective of chemistry or degree of interpenetration; some small variations in the positions of metastable phases (local minima) are observed.

I am entirely sympathetic to the motivation for this study. It is clearly very well carried out, and the paper is well written (in particular it is beautifully illustrated).

We would like to thank the reviewer for their careful reading of the manuscript and their constructive remarks, which have helped to substantially improve the manuscript. To this end we have performed a substantial amount of new calculations to also include water loaded frameworks and evaluate the effect hereof on the observed phase. More details are given below in the specific responses to the referee's comments. Overall, we have accounted for all comments formulated by the reviewer, and revised the manuscript and Supporting Information accordingly. In what follows, detailed replies to the various points raised are given.

1. Nevertheless, I don't think that the study can be published in Communications Chemistry because it fails to reproduce or rationalise the experimental flexibility of e.g. COF-300. [... (see remark 2)] As such, the study can only have limited impact within its immediate field. Probably the real value of the paper is to set out a methodology for exploring flexibility in diamondoid frameworks – this might be suitable for a society journal, for example. But even in that case it would have been satisfying to see the authors engage with the difference between calculation and experiment to uncover the underlying reason for the discrepancy between the two.

Large-scale flexibility in **dia** COFs is up to this point only observed under the adsorption of either water or organic molecules (while ref. [1] reports on the "temperature-induced" flexibility of COF-300, the authors explain that the observed flexibility is triggered by the release of water, making it rather a guest-induced transition). As detailed in Table S1, single-crystal X-ray diffraction (SXRD), powder X-ray diffraction (PXRD), and electron diffraction tomography (EDT) studies indicate that COF-300(7) contracts upon water adsorption [2-4] and expands upon the adsorption of either poly(methyl methacrylate) (PMMA) or tetrahydrofuran (THF) [3,4]. All these transitions retain the square shape of the pore, and only lead to variations in the size of this square-like pore. In terms of our collective variables (CVs) $CV_1 \times CV_2$, water adsorption contracts the pore from either 13.1 Å x 13.1 Å (SXRD) or 10.2 Å x 10.2 Å (PXRD and EDT) to 9.8-9.9 Å x 9.8-9.9 Å, whereas both PMMA and THF expand the pore to 13.6-13.7 Å x 13.6-13.7 Å. At this point, three main observations can be made:

- i. Our original manuscript only reported on the effect of temperature and degree of interpenetration as triggers for the occurrence of different (meta)stable states, not on the effect of gas adsorption. Hence, the observation that the SQ phase is always the most stable

phase in Figure 4 and 5 is not in disagreement with the experimentally observed flexibility upon adsorption.

- ii. The size variations observed experimentally upon adsorption (see above) are substantially smaller than the difference between the SQ phase (15.9 Å x 15.9 Å) and the sq phase (6.6 Å x 6.6 Å) reported in Figure 4 (for COF-300(3)). Hence, the potential flexibility by switching between the sq and SQ phases revealed in Figure 4 is larger than any of the size variations observed experimentally for this material.
- iii. The experimentally observed SQ volume of the empty COF-300(7) framework differs between the SXRD study on the one hand and the PXRD and EDT studies on the other, which may be due to the different synthesis protocols.

However, we fully understand the concern of the referee regarding the connection of the computational results with the experimental observations. Therefore, we have performed a series of extra simulations to complement our original observations, probing the response of **dia** COFs upon adsorption. More in particular, we modelled the water-induced changes in the phase stability for COF-300(7). To this end we used the well validated TIP4P force field for water. However some care needs to be taken in the description of the noncovalent interactions, as the noncovalent framework-framework interactions were based on a different functional form, and thus parameterization, with respect to the TIP4P force field (MM3-Buckingham vs. Lennard-Jones). As such, individual Lennard-Jones cross terms were introduced, as discussed in the new Section S6.5.

Through performing simulations on the COF-300(7) framework for a range of different water loadings, free energy surfaces were generated. They predict the variation in the framework volume and in the collective variables as a function of the water loading, as outlined in the new paragraph at the end of the Results section, supplemented by the new Section S5.2 in the Supporting Information. As reproduced in Figure 1, a loading of eight water molecules per channel in the unit cell gives rise to the largest volumetric change, contracting the pore (expressed as a function of the CVs) from 13.9 Å x 14.1

Figure 1: Free energy surfaces for COF-300(7) at 300 K as a function of water loading, exponentially varied from 1 to 16, as indicated in the top left corner of each subpanel. The different (meta)stable phases are indicated on the figure. Regions with a free energy exceeding the free energy minimum by more than 500 kJ/mol are omitted from the plot, as they are not accessible under realistic conditions. The final subpanel shows the exact position of the SQ phase for each loading in detail.

Å to 13.1 Å x 13.6 Å and the unit cell volume from 5594 Å³ to 5275 Å³. While these values differ somewhat from those observed experimentally, the relative change in volume of 6 % upon water adsorption is very similar to the value obtained in the experimental studies mentioned above. While the agreement is not perfect, we conclude that our protocol and force field succeed in qualitatively describing the experimental pore shrinkage upon water adsorption. A more detailed investigation of the structure with guest adsorption would require an extensive study on the sensitivity of the force fields which is beyond the scope of the current manuscript.

We have taken up all water-loaded COF-300(7) free energy surfaces in a new paragraph at the end of the Results section, which reads:

“Finally, while it is clear that the degree of interpenetration and the temperature impart significant variations to the transition barriers, the SQ phase remains the most stable phase for all considered materials. However, as reported in Table S1, various channel shapes emerge in diamondoid COFs when exposed to different guest molecules, such as water. As such, we herein sought to explore how the inclusion of water impacts the structure and relative stability of the different phases in COF-300(7), which experimentally amounts to a volume contraction of about 5% for the SQ phase.^{26–28} Similar to Figure 4, the effect of water loading is observed through analysing the variations in the free energy landscapes, when increasing the loading from one to sixteen water molecules per channel. This is illustrated in Figure 6, where the loading indicated in the top left corner of each pane. From the final panel, it is clear that the SQ channel first decreases in size when water is present in the pores, with a minimum volume for a loading of eight molecules per channel, after which the size increases again. Additionally, a decrease of the available phase space can be observed, due to the density increase, similar to the interpenetration effect. When comparing the most contracted water-filled framework to the guest-free framework, the contraction corresponds to a change of (expressed as a function of the collective variables) 13.9 Å x 14.1 Å to 13.1 Å x 13.6 Å, or expressed in terms of the unit cell volume of circa 5594 Å³ to circa 5275 Å³ (–6%). While these values differ somewhat from those observed experimentally, as tabulated in Table S6, the relative change in volume upon water adsorption is very similar to the values from those references mentioned in Table S1, succeeding in a qualitative description of the experimentally observed pore shrinkage upon water adsorption. This adsorption-induced shrinkage is substantially smaller than the potential temperature-induced flexibility between the SQ and sq phases for the empty framework predicted in Figure 4.”

We also added the following sentence to the discussion (page 15):

“However, by varying the temperature, we observed that the temperature cannot be straightforwardly used as a driving force to completely destabilise the SQ phase. Instead, through introducing water molecules, significant size variations could be induced, shifting the SQ phase towards lower volumes.”

The simulations discussed above required a non-trivial extension of the noncovalent part of our original COF-300 force field to model the interactions between the framework and our existing TIP4P-based water force field from ref. [5]. For consistency, we also updated the free energy surfaces of the empty frameworks throughout the main manuscript and the Supporting Information, although we emphasise that no major changes were observed, and all conclusions drawn earlier remain valid.

2. I do not see that the key claim of page 9 “...the relative stability of the distinct phases is indeed substantially impacted depending on the interpenetration” is actually supported by the data of Figure 4: the sq phase is always the most stable; the same is true with variation in chemistry or temperature (Figure 5).

Although the SQ phase remains the most stable phase throughout the different degrees of interpenetration reported in Figure 4 and throughout the different degrees of interpenetration and temperatures reported in Figure 5, the following observations do substantiate our claim in our opinion:

- i. Depending on the degree of interpenetration, the sq and rect phases appear as metastable phases next to the stable SQ phase.
- ii. The relative stability of the sq and rect phases with respect to the stable SQ phase, as well as the transition barriers between them, change by several hundreds of kJ/mol depending on the temperature and degree of interpenetration, as shown in Figure 5.
- iii. The size and shape variations that accompany transitions from the SQ phase to either the rect or sq phase are larger than the expansion and shrinkage of the SQ phase observed under adsorption (see point 1 above). Therefore, knowledge that these two phases can be stabilised is important to fully exploit the flexibility in these materials.
- iv. There is a significant difference between the free energy landscapes as a function of interpenetration for the flexible materials (COF-300, COF-320) compared to the rigid materials (NPN-1, NPN-3), which motivated our conclusion on page 14 that “certain structural decorations preclude flexibility, such as the rigid azodioxy linker and adamantane cage in the NPN COFs, [whereas] others, such as the degree of interpenetration can fundamentally shift the delicate equilibrium between the stabilisation and destabilisation effects, driving the phase equilibrium.”

3. Wherever the authors choose to send on this manuscript, one point I would encourage them to take into account is to tone down their claims regarding the universality of their description of dia phases. There are many additional symmetry-breaking distortions - e.g. shear, loss of edge transitivity, orthogonal tilts - that are not taken into account here.

We acknowledge that the generality of our representation of the phase landscape for diamondoid COFs is indeed limited to describe the specific size and shape variations described in Figure 2 and in this sense we have formulated the statements in the manuscript in a more balanced way, which clearly emphasize the potential importance of other more complex, material distortions. However the simulation of such effects would clearly be beyond the scope of this manuscript as completely other models accounting for a much higher level of complexity would be necessary for such simulations. The revised statement on page 4 now reads:

“When viewed along this axis, the material exhibits large square-shaped (SQ) channels, independent of the degree of interpenetration, as visualised in Figure 2c. Experiments have shown that, beside these SQ channels, the channel network in diamondoid COFs can adopt various sizes and shapes (see Figure 2c), either due to different synthesis protocols or the presence of guest molecules. This gives rise to smaller square-shaped channels (sq) or rectangular channels (rect), respectively, as reported in Table S1 in the Supporting Information. Aside from these variations, other, possibly more complex, material distortions can manifest depending on the external stimuli, such as a collective shearing or individual tilting of the tetratopic nodes.^{35,36} However, as these additional material distortions have not yet been associated with flexible diamondoid COFs experimentally, and accounting for their

contributions to the phase stability would require substantially more complex models, they are not explicitly taken into account in this work. As such, in general, three distinct channel shapes can be identified in the diamondoid topology, namely, SQ, sq and rect.”

Reference 35,36 in the introduced text refer to appropriate descriptions of typical material distortions in *e.g.* perovskites.

REVIEWER #2

This manuscript by Van Speybroeck and co-workers presents the Free Energy Surfaces of four related diamondoid COFs: COF-300, COF-320, NPN-1 and NPN-3 and discusses the availability of structural transitions between large square pore, small square pore and rectangle pore geometries for different degrees of interpenetration, across the four COFs (two nodes, three linkers).

Clearly, the authors have put quite some thought into the presentation of their results, especially their figures, which I commend them for. My comments are therefore minor.

We would like to thank the reviewer for their careful reading of the manuscript, the very positive evaluation and their constructive remarks. We have considered the comments raised by the reviewer, and improved the manuscript and Supporting Information accordingly. In what follows, detailed answers to all comments are given.

There are two more things I'd like to see in the SI:

- 1. In Table S1, for the filled squares, it would be nice to see the guest molecule(s) named as the extent and type of interaction with the framework would be expected to affect the observed COF structure.**

We agree with the reviewer and have added the guest molecule for the appropriate entries in Table S1 on page S-7.

- 2. The second thing that would be nice is a table in the SI (probably linked to the github) that links the two CVs (useful to computational colleagues) to lattice vectors and pore size (more recognizable to experimental colleagues) and then the structures themselves on github (I found the initial structures, not the optimized ones).**

We added a new Section S3 "Correlation between collective variables and system properties" to the Supporting Information to provide a clear picture of the relation between the collective variables and the proposed system properties. The main conclusions from this comparison in the Supporting Information read:

"Notably, the asymmetry in the collective variables is not reflected in the cell parameters. This follows from the fact that within one unit cell four channels are captured, which cancel each other out, as illustrated in Figure S1. Evidently, the pore volume is strongly correlated with channel size changes, as defined in the main text. However, shape variations also appear to have an indirect effect on the pore volume. This can be attributed to the changes in the elongation of the diamondoid cage, i.e. to variations of the *c* parameter. As already discussed in the main text, size decreases effectively correspond to an elongation of the diamondoid cage. However, when a shape change occurs, the nature of the steric interactions between the tetratopic building units changes. This results in a smaller elongation of the diamondoid cage for the same size, lowering the pore volume."

For all initial structures on our GitHub (which are collected in the folder *initial_structures*), the corresponding optimised structures have been added as well. Moreover, an additional folder *CV_measurement* was added in which the CIF information and the scripts required to calculate the CV values have been provided for each material of Table S1. Moreover, a subfolder *link_CV_pore* was added, which contains the scripts to reproduce the new Figure S2. Finally, a subfolder *generate_CV*

was added and contains a script to perform an umbrella sampling simulation, to drive the system towards certain predefined CV values, after which an optimization might result in a (meta)stable geometry for that CV combination.

3. Typos:

Page 7: There is a reference to "Section ??" - obviously something didn't resolve.

Page 9: "buildin" blocks

Page S-4: "interpenetration" axis

These typos have been corrected in the main text and the Supporting Information.

REVIEWER #3

The paper by Borgmans et al. reports a theoretical investigation of different diamondoid COF networks concerning their flexibility to undergo pressure or temperature-induced phase transformations. The corresponding “breathing” effect for the related crystalline porous frameworks from coordination compounds or MOFs has attracted a lot of interest and research activity. As the authors correctly point out, this is not so for the covalent organic frameworks with their potentially “stiffer” bonds. However, also the 3D COFs based on the dia topology can undergo a transformation leading to reduced cell volumes. Since COFs have their advantages over MOFs in particular applications, such a study is interesting and relevant for the entire field.

The authors have used their QuickFF strategy to train a molecular mechanics potential wrt. to DFT reference data for the particular system. They investigated different COFs, varying the linker length and rigidity. Most importantly, the dia topology is prone to interpenetration and the results were thus compared for different degrees of interpenetration. Free energies for deformation along the two relevant deformations towards an isotropically reduced cell (SQ \rightarrow sq) and a deformed system with rectangular pores (SQ \rightarrow rect) via a rotation of the 4 connected vertices were computed.

The paper is very well-readable and clear. The viewgraphs are exceptionally well done and help in understanding the approach. I also find the definition and discussion of the collective variables for the above-mentioned deformations of the MOFs smart and easy to follow.

Overall, this is a very interesting paper and I recommend its publication as is (on page 7 the authors should fix the missing reference (Section ??)).

We would like to thank the reviewer for their careful reading of the manuscript and the very positive evaluation. We have fixed the missing reference on page 7 in the revised version.

References

- [1] Moroni, M.; Roldan-Molina, E.; Vismara, R.; Galli, S. & Navarro, J. A. R., Impact of Pore Flexibility in Imine-Linked Covalent Organic Frameworks on Benzene and Cyclohexane Adsorption. *ACS Appl. Mater. Interfaces*, **2022**, 14, 40890-40901,
- [2] Yang, Q.-Y.; Lama, P.; Sen, S.; Lusi, M.; Chen, K.-J.; Gao, W.-Y.; Shivanna, M.; Pham, T.; Hosono, N.; Kusaka, S.; Perry, J. J.; Ma, S.; Space, B.; Barbour, L. J.; Kitagawa, S.; Zaworotko, M. J. Reversible Switching between Highly Porous and Nonporous Phases of an Interpenetrated Diamondoid Coordination Network That Exhibits Gate-Opening at Methane Storage Pressures. *Angew. Chem. Int. Ed.*, **2018**, 57, 5684–5689.
- [3] Ma, T.; Kapustin, E. A.; Yin, S. X.; Liang, L.; Zhou, Z.; Niu, J.; Li, L.-H.; Wang, Y.; Su, J.; Li, J.; Wang, X.; Wang, W. D.; Wang, W.; Sun, J.; Yaghi, O. M. Single-crystal x-ray diffraction structures of covalent organic frameworks. *Science*, **2018**, 361, 48–52.
- [4] Chen, Y.; Shi, Z.-L.; Wei, L.; Zhou, B.; Tan, J.; Zhou, H.-L.; Zhang, Y.-B. Guest-Dependent Dynamics in a 3D Covalent Organic Framework. *J. Am. Chem. Soc.*, **2019**, 141, 3298–3303.
- [5] Sun, Y.; Rogge, S. M. J.; Lamaire, A.; Vandenbrande, S.; Wieme, J.; Siviour, C. R.; Speybroeck, V. V.; Tan, J.-C. High-rate nanofluidic energy absorption in porous zeolitic frameworks. *Nat. Mater.*, **2021**, 20, 1015–1023.

REVIEWERS' COMMENTS:

Reviewer #1 (Remarks to the Author):

Editorial note: this reviewer provided no further comments for the authors.

Reviewer #2 (Remarks to the Author):

The authors have addressed the comments of all three reviewers carefully and admirably.

One tiny request: In the new Figure 6, the blue spot (\Rightarrow 2H₂O) is not visible, I assume it is coincident with another point. Perhaps a 50/50 dot would make this clear?